# Does Pain Acceptance Contribute to Improved Functionality through Walking in Women with Fibromyalgia? Looking at Depressive Comorbidity

**DOI:** 10.3390/ijerph20065005

**Published:** 2023-03-12

**Authors:** Cecilia Peñacoba, Carmen Ecija, Lorena Gutiérrez, Patricia Catalá

**Affiliations:** Departamento de Psicología, Facultad de Ciencias de la Salud, Universidad Rey Juan Carlos, 28922 Madrid, Spain

**Keywords:** fibromyalgia, functional limitation, acceptance, depression, chronic pain, comorbidity

## Abstract

In the last decade, research has pointed to physical exercise as an effective treatment in fibromyalgia patients. Some studies have highlighted the role of acceptance and commitment therapy in optimizing the benefits of exercise in patients. However, given the high comorbidity in fibromyalgia, it is necessary to value its possible influence on the effect of certain variables, such as acceptance, on the benefits of treatments, such as physical exercise. Our aim is to test the role of acceptance in the benefits of walking over functional limitation, further assessing whether this model is equally valid, considering depressive symptomatology as an additional differential diagnosis. A cross-sectional study with a convenience sample through contacting Spanish fibromyalgia associations was carried out. A total of 231 women with fibromyalgia (mean age 56.91 years) participated in the study. Data were analyzed with the Process program (Model 4, Model 58, Model 7). The results highlight the role of acceptance as a mediator between walking and functional limitation (B = −1.86, SE = 0.93, 95% CI = [−3.83, −0.15]). This model, when depression is incorporated as a moderator, is significant only in patients without depression, revealing the need for personalized treatments in fibromyalgia, considering their most prevalent comorbidity.

## 1. Introduction

Recent studies have considered fibromyalgia (FM) as a central sensitization syndrome, since they suggest that the predominant pathogenic mechanism is the alteration of pain regulation at the brain level [1]. Traditionally, pain management paradigms have aimed on one hand to decrease pain levels and on the other to increase control over pain responses [2]. In contrast, the effects of these interventions have not always been as desired [3]. In recent times, health research has recommended multidisciplinary treatments for fibromyalgia patients, which combine psycho-education, recommendations to carry out a healthy lifestyle, and non-pharmacological interventions [4]. Physical exercise together with psychological therapy is considered the most effective treatment for fibromyalgia [3].

Within the different modalities of aerobic physical exercise, it has been proven that regular walking offers multiple benefits in these patients [5,6]. Some of the most notable benefits are the reduction of the impact of the disease on daily life, fatigue or improvement in functional limitation, and pain control [7,8]. However, walking with the aim of reducing pain levels can be counterproductive if the way of coping with the activity is not adequate (e.g., pain avoidance) [9]. When pain is interpreted as a threat, it leads to kinesophobia, defined as excessive fear of movement or fear of physical activity, which can induce disability [10]. Despite the positive effects of walking as a form of physical exercise, the results are not always as desired, given the low adherence that patients with fibromyalgia have to this healthy behavior [11]. For this reason, previous researchers have focused their interest on analyzing the risk factors for non-adherence, the so-called inhibitors of walking behavior. In this sense, the symptoms of fibromyalgia, pain, and fatigue are shown as the most frequent inhibitors, causing the perception and coping that the patient makes of them to be especially relevant [12]. Pain catastrophizing has been analyzed within motivational approaches, which understand it as the result of the conflict of goals: goals aimed at pain control (basically through activity avoidance behaviors) and goals with a vital purpose for patients (normally associated with carrying out an activity) [13]. There is a large amount of research that has focused on the role of catastrophizing as a maladaptive filter for interpreting symptoms and their future consequences. Fibromyalgia patients with high catastrophizing scores perform less physical activity and present greater pain and fatigue and greater functional limitation [14,15].

Therefore, effective coping is the key to control the symptoms and improve the quality of life of these patients, especially when a treatment potentially associated with experiencing symptoms such as pain and fatigue (i.e., walking) is prescribed. In this sense, compared to the abundant research on catastrophizing as maladaptive coping, as we have pointed out, research on adaptive strategies is considerably less. The latest research shows that the benefits of pain acceptance go beyond standard coping strategies [2]. In a recent study comparing both pain catastrophizing and acceptance in fibromyalgia and obese patients, an association was found between lower levels of activity (both self-reported and through the 6-min walking test (6MWT)) with higher pain catastrophizing and lower acceptance scores [16]. A current study noted that pain acceptance explained six times the variance in seven measures of physical and psychosocial functioning compared to five widely studied coping strategies [17]. The acceptance of pain not only involves learning to live with ongoing pain without trying to avoid, change, or reduce it, but also involves making conscious decisions about the prescribed treatment (i.e., walking) [18,19]. In addition, the acceptance process includes recognizing that pain is an integral part of fibromyalgia and that, despite there being no cure, one can learn to manage pain effectively. There is evidence that people who have a greater acceptance of pain are free to pay attention to it frequently. This fact allows them, on the one hand, to have a greater capacity, motivation, and commitment to carry out an active lifestyle and, on the other, to maintain a positive vision of life [2]. Likewise, previous studies have pointed to this ability as a powerful correlate of positive physical, psychological, and social adjustment to chronic pain [20,21,22]. There are data that confirm that a greater acceptance is related to less intensity of pain, symptoms, and disability and to better physical functionality [3,23,24]. Despite this, the role of pain acceptance in fibromyalgia patients remains controversial. The challenge of the most current research lies in revealing what factors are influencing the results [3].

Knowing the heterogeneity among patients with fibromyalgia, due in large part to the complexity of the disorder and the high associated comorbidity [25], it is possible that the mechanisms involved do not work the same in all patients [26]. In fact, different studies have pointed out the need to identify subgroups within the heterogeneity of patients with fibromyalgia in order to rationalize and clearly define the most appropriate interventions [27,28]. Given the broad comorbidity associated with fibromyalgia, it is not surprising that the symptoms themselves formed the basis for establishing subgroups [29]. Thus, by way of example, Pérez-Aranda et al. [29] use the FIQR as a general measure of the impact of the disease, establishing four differential subgroups that allow the detection of differences for economic costs and for different clinical outcomes (i.e., anxiety, depression, stress, cognitive impairment, inflammatory markers’ levels, gray matter volumes). Vicent et al. [30], using the usual symptomatology in patients with fibromyalgia (i.e., pain, fatigue, function, sleep disturbance, depression, dyscognition, anxiety, stiffness) as a criterion, found four clusters that classify the sample in severity levels: Clusters 1 and 4 refer to the lowest and highest average levels across all symptoms, respectively, and clusters 2 and 3 reflect moderate symptoms levels, the latter differing in the severity of the depressive and anxiety symptoms.

Increasing data support the comorbidity of fibromyalgia and psychiatric conditions [31,32]. A recent review indicates that depression/major depressive disorder is considered the most prevalent psychiatric comorbidity in this population, noting that more than half of the patients with fibromyalgia (weighted prevalence up to 63%) have received this diagnosis throughout their lives. However, anxiety-related disorders were much less common [32]. The data from this review point to the need to consider depression as a comorbid disorder that is highly present in patients with fibromyalgia, to increase the generalization and applicability of the research results [32]. In order to shed some light on the influence of depression in patients with fibromyalgia and in particular on the processes involved in adherence to walking, the present research aims first to evaluate the mediating role of pain acceptance as a key variable in the benefits of walking behavior on functional limitation, and second to verify the validity of this model when considering depression as a comorbidity. The proposed model is really based on two conceptual approaches: (a) On the one hand, in relation to pain acceptance as a mediator, the application of acceptance and commitment therapy (ACT) in patients with chronic pain shows the positive results of the same in the acceptance of their chronic condition and in the increase in functional autonomy [33]. In this context, acceptance is a nuclear variable of the model. The process of acceptance of chronic pain is associated with a lower impact of the disease, including less disability [34,35]. Specifically, changes in acceptance, pain-related anxiety, compensation strategies, and pain interference in walking ability have been found after the application of ACT in chronic pain [36]. (b) On the other hand, proposing depression as a moderating variable in the model is due to the heterogeneity previously described in fibromyalgia [27] and the need to establish profiles or subgroups to respond to this reality and design more personalized treatments. Given the prevalence of depressive comorbidity in patients with fibromyalgia [32], the analysis of this variable is proposed, establishing subgroups and incorporating depression as a moderating variable. Taking into account the existing literature, it is hypothesized that acceptance benefits the relationship between walking behavior and functional limitation and that this mechanism varies depending on the presence or absence of depression.

## 2. Materials and Methods

### 2.1. Participants

A total of 268 participants agreed to participate in the study. All of them met the following inclusion criteria: being a woman, being older than 18 years, and having received a diagnosis of fibromyalgia by rheumatologists or primary care physicians. All of them were diagnosed according to the American College of Rheumatology criteria [37]. As exclusion criteria, the existence of concomitant rheumatic disorders and the existence of psychotic disorders were taken into account. A minimum *n* of 200 was established, following the criteria established for the regression analyses [38], and the recommendations for the analyzes moderation using the PROCESS tool in SPSS [39]. Finally, effective responses were obtained from 231 patients (22 patients did not attend the scheduled evaluation appointment, 6 patients did not sign the informed consent, and the questionnaires of 9 patients contained a large amount of missing data, so it was decided to eliminate them from the study). All the women were recruited from different mutual aid associations in Spain. In Spain, most patients with fibromyalgia belong to an association [40]. Mutual aid associations represent significant savings for both patients and the health system, which is why it is a very common practice [33]. A psychologist from the research team went to the different associations to deliver an evaluation protocol to the patients. Completing this protocol lasted between 20 and 30 min. Ethical principles for research with human participants were followed for all evaluation procedures. The University Ethics Committee (Universidad Rey Juan Carlos, Reference number: PI17/00858) approved this study.

### 2.2. Measures

Walking: We used an ad hoc dichotomous question (no = 0/yes = 1) to test whether participants engaged in the behavior of walking for physical exercise. Specifically, one of the walking patterns usually recommended for patients with fibromyalgia was evaluated: “walk 2 to 4 days a week, a minimum of 30 min a day, in 15–20 min shifts, with a small rest between shifts for a minimum of six consecutive weeks in order to exercise” [41].

Acceptance: The Chronic Pain Acceptance Questionnaire (CPAQ) [34] is a self-report questionnaire designed to assess acceptance of chronic pain. In this study, the validation into Spanish was used [42]. The CPAQ questionnaire is made up of 20 items that present a bifactorial structure: pain willingness (11 items) and activities engagement (9 items). The response scale used is 7 Likert-type points, where 0 is never true and 6 is always true. The total score of the scale ranges from 0 to 66. In this study we used the pain willingness subscale. Higher scores mean a high acceptance of pain. The validity and reliability of this instrument has been demonstrated [42]. In our sample, Cronbach’s alpha for the pain willingness subscale was 0.84.

Functional limitation: The Revised Fibromyalgia Impact Questionnaire (FIQ-R) [43] is a 21-item self-report questionnaire that assesses three factors: physical function, general impact, and symptoms. For this study, the functional limitation subscale of the Spanish version of FIQ-R [44] was used, which evaluates the degree of difficulty experienced by the patients when performing a series of physical exercises in activities of daily living (for example, “going shopping” or “climbing stairs”). A Likert-type response format of 11 points ranging from 0 to 10 is used. To obtain the score for this subscale, the first nine items are added and divided by three. Higher scores indicate less functionality. The subscale has obtained good reliability and validity indices in previous studies [45]. Cronbach’s alpha in the present study was 0.88.

Depression: The medical history was examined to verify a diagnosis of depression by a psychiatric professional. In addition, this diagnosis was verified through the administration of the Spanish version of the Hospital Anxiety and Depression Scale (HADS) [46,47], depression dimension, establishing a cut-off point score of 12 or higher in this population [48].

Pain intensity: The Brief Pain Inventory is an instrument used to assess pain intensity [49]. In this study, the average of the four items (BPI items) that make up the questionnaire (maximum, minimum, and general pain intensity during the last 7 days and current pain intensity) was used. The response scale ranges from 0 (no pain) to 10 (the worst pain imaginable). This procedure for measuring pain intensity has been widely used in the pain literature [50]. In this study, the internal consistency of this scale was high (α = 0.86).

Sociodemographic and clinical data: Questions related to age, marital status, educational level, employment status, medication prescribed, and time since diagnosis of the participants were included.

### 2.3. Data Analysis

For data analysis, the statistical package SPSS 22 [51] was used. The bivariate associations between the variables under study (walking, depression, acceptance, and functional limitation) were analyzed and then a series of multivariate regressions were calculated through the macro PROCESS [52]. Specifically, a mediation analysis was performed with Model 4 and a moderate mediation analysis with Model 58. Acceptance was used as a mediator, depression as a moderator, walking as an independent variable, and functional limitation as the outcome. Pain intensity was included as a covariate for all models. Statistical significance was set at an alpha level of 0.05. The PROCESS macro uses the ordinary least squares (OLS) analysis to calculate the mediation and moderate mediation effects, and the bootstrap is used to calculate the confidence intervals (CI). Specifically, bias-corrected bootstrap CIs based on 5000 bootstrap samples with a 95% confidence level are used. When the confidence intervals do not include zero, the effect is considered significant. Non-centered variables were used in the post hoc analyses in order to facilitate the interpretation of the results.

## 3. Results

### 3.1. Sample Characteristics

The age of the participants ranged from 19 to 78 years (mean = 53.89; SD = 9.25). Seventy-five percent of the women were married or in a stable relationship. The remaining 25% were distributed in the following categories: separated or divorced (12%), single (8%), and widows (5%). Most of the participants had completed upper secondary education (46.3%). Similar percentages were found for university studies (24%) and for primary education (27%). Only 2.7% had not completed official studies, only knowing how to read and write. Regarding the medication they took on a daily basis, 16 women took antidepressants (21%), 22 women took anti-inflammatory drugs (24.3%), and 38 women took other types of medication to regulate their glucose levels, blood pressure, or allergic problems (54.7%). The mean time since diagnosis of fibromyalgia was 9.85 years (SD = 8.49; range 1–46 years). Mean pain intensity was 7.05 (SD = 1.49; range 1–10).

### 3.2. Descriptive Analysis and Correlations

Descriptive data for walking, depression, acceptance, and functional limitation are shown in Table 1. Regarding the correlations between the variables under study, acceptance was negatively correlated with functional limitation (*p* < 0.001). In addition, significant differences were observed in patients with depression versus without depression in acceptance (t = 6.93, *p* < 0.001) and functional limitation (t = −4.84, *p* < 0.001). Patients without depression obtained higher scores in acceptance and lower scores in functional limitation. Similarly, significant differences were observed in patients who walked versus those who did not walk in acceptance (t = −2.14, *p* = 0.036) and functional limitation (t = 3.53, *p* < 0.001). The patients who walked obtained higher scores in acceptance and lower scores in functional limitation. Finally, significant differences were observed in patients with depression versus without depression and those who walked versus did not walk (X^2^ = 8.78, *p* = 0.002). Specifically, 65.9% of patients without depression walk, while 46.1% of patients with depression do so.

### 3.3. Mediation Model of the Relationship between Walking and Functional Limitation with Acceptance as a Mediator

Figure 1 shows the results of the mediation analysis controlling for the effects of pain intensity. Acceptance fully mediates the effect of walking on functional limitation. The total model effect was significant (B = −1.86, SE = 0.93, 95% CI = [−3.83, −0.15]). In addition, there was a direct effect of walking on functional limitation (B = −5.87, SE = 2.09, t = −2.80, 95% CI = [−10.00, −1.74], *p* = 0.05). The mediation model explains a total variance of 46% (F = 64.20, *p* < 0.001).

### 3.4. Moderate Mediation Model

Table 2 shows results of the moderate mediation model (Model 58). This model (see Figure 2) includes a mediation process where acceptance is the mediating variable between walking (predictor) and functional limitation (outcome). Additionally, the model includes the possible moderating role of depression between walking and acceptance on the one hand, and between acceptance and functional limitation on the other hand. Pain is entered as a covariate in the model. The results indicate a significant indirect effect only for the interaction between walking and depression, as the interaction between acceptance and depression was not significant. That is, we follow this model with a simpler model (Model 7). The indirect effect of walking that predicts functional limitation through acceptance was conditioned by the presence of depression (moderate mediation index = 2.82, SE = 1.47, 95% CI [0.157, 5.978]). The results reveal that depression conditions the contribution of walking to acceptance. Specifically, this relationship (walking to acceptance) was significant in patients without a diagnosis of depression (B = −2.05, SE = 1.11, 95% CI [−4.53, −0.22]) (Table 3). This indicates that the positive effect of walking on functional limitation through acceptance is favored when patients do not present depression. The proposed model contributes to the explanation of 49% of the variance of the functional limitation. 

## 4. Discussion

This research analyzed the mediating role of acceptance in the relationship between walking behavior and functional limitation and tried to verify if the relationship established between walking behavior and functional limitation through acceptance is maintained when it takes into account whether or not patients present depression. Considering the first aim, the results show that the effect of walking on functional limitation is completely mediated by acceptance. According to acceptance and commitment therapy, acceptance not only implies accepting pain passively, but also makes it easier to carry out valuable activities for the person without the need to avoid or control pain [53]. It has been shown that patients who accept their experiences are more willing to undertake efforts to relieve their pain [54] such as going for regular walks. In this context, it has also been pointed out that the acceptance of pain allows patients to better adapt to suffering from chronic pain [55] and to maintain adaptive daily functioning while continuing to experience pain [34]. Taking into account what is stated here and the model proposed, it could be said that patients who manage to accept pain as an integral part of fibromyalgia may perceive an improvement in their ability to carry out daily activities by incorporating regular physical activity into their lifestyle. Therefore, for treatment based on physical activity to have a beneficial impact on the health of these patients, it must be combined with psychological techniques for pain acceptance. These results could even point to the need to consider acceptance as a differentiating feature when establishing subgroups in patients with fibromyalgia. As we have previously pointed out, most of the preceding research establishes subgroups of patients based on symptomatology [39]. However, some research also includes personality traits and cognitive–emotional variables as factors of interest when establishing differential profiles. Specifically, pain catastrophism is included in some of these clusters [56,57]. To our knowledge, the role of acceptance has not been included when establishing differential profiles within patients with fibromyalgia. Likewise, it would be interesting to assess the role of kinesophobia, since the fear of movement could also influence the ability to walk [10].

In line with the second aim, when performing the moderate mediation model, the findings showed that the presence of depression modifies the effect of walking on acceptance. Specifically, the patients perceive the benefits of walking on functional limitation through the acceptance of pain only when they do not present a diagnosis of depression. Previous studies had already reported the beneficial role of positive health factors in the symptoms of fibromyalgia [58,59,60]. These results show the need to identify subgroups of patients with fibromyalgia in order to carry out adequate treatments. Previous studies already mentioned the usefulness of establishing subsets based on symptomatology and biopsychosocial factors [57,61,62]. In this line, different subgroups derived from exploratory cluster analyses of a wide range of variables have been published [29,62,63]. For the most part, studies have established subgroups based on clinical characteristics [29,63,64]. Studies that analyze having depression as a differential diagnosis in fibromyalgia have pointed out the influence that this has on the health outcomes of these patients. Specifically, it has been observed that fibromyalgia patients with depression have lower scores on positive affect, higher scores on pain vigilance and negative affect, and slower reaction times than FM patients with low depression and pain-free controls [65]. In studies in which anxiety and depression are studied jointly as comorbid symptoms in FM patients, compared with fibromyalgia patients without those diagnoses, it has been verified that the main effects of anxiety and depression were significant for the index scores on activity-related discomfort, subjective work capacity, and quality of life [66]. However, it is important to note that in the latter case, it would be interesting to study depression independently, since different studies have reported that anxiety and depression are independently associated with the severity of pain symptoms in fibromyalgia [67,68], and that the proportions of comorbid anxiety and depression are different. It has been proven that the proportion of patients with fibromyalgia and comorbid depression is much higher than with anxiety [68]. However, to our knowledge, until now, the mediating effect of acceptance in the relationship between walking behavior and limitation in fibromyalgia patients with and without depression had not been evaluated. Despite having results on the role of acceptance in adherence to the behavior of walking and its benefits [69], in view of our results it seems that this role of acceptance does not work in the same way in patients with depressive comorbidity. A recent review that analyzes the effectiveness of the interventions in patients with fibromyalgia of the therapy of acceptance commitment and full attention already indicated that in patients with depression, the effects are small [24]. It is possible that the consequences of suffering from depression are influencing these results. Patients with this disorder are known to experience anhedonia, negative thoughts, lack of motivation, exhaustion, hopelessness, or a feeling that things will never get better, which can make it more difficult for a person to find ways to manage pain. In addition, it has been proven that depression can exacerbate physical pain and cause fatigue and a feeling of exhaustion, which could make it even more difficult for these patients to find benefits in carrying out the behavior of walking [70,71].

These findings have important practical implications as they suggest that implementation-oriented treatments for walking in fibromyalgia do not have the same effect in patients with depression. In this last case, a priori, walking would not have a positive impact on the functionality of patients with fibromyalgia. Thus, working in the first place to reduce the levels of depression in these patients would be the most appropriate treatment. These results are consistent with previous studies showing that the benefits of leading an active lifestyle vary depending on the characteristics of the population examined [12,72,73]. Our results confirm that, before promoting the pre-registration of aerobic physical exercise (e.g., walking) in order to increase the functionality of these patients, it would be necessary to carry out a previous analysis of the levels of depression.

One of the limitations of this study was its design; given its cross-sectional nature it is not possible to establish cause–effect relationships. In addition, the population was limited to women only, which makes it difficult to generalize the results, despite women being the predominant gender in the diagnosis of fibromyalgia. Therefore, it is considered necessary to carry out more research in other populations with chronic pain and men. Finally, self-report questionnaires were used for most measures. Despite being a common problem in this area in the existing literature [74], this could affect the results.

## 5. Conclusions

In conclusion, the findings presented here are relevant both for the field of research and for the clinical setting. Specifically, the results point to acceptance as a relevant psychological mechanism to perceive the benefits of carrying out an active lifestyle (e.g., going for a walk) on functional limitation. However, considering depression as a comorbidity, the validity of this model varies, being significant in patients without depressive comorbidity. Thus, it appears that acceptance is a key positive marker, especially when patients do not report depression. This is important to increase the efficacy of treatments in patients with multiple health conditions comorbid with fibromyalgia.

## Figures and Tables

**Figure 1 ijerph-20-05005-f001:**
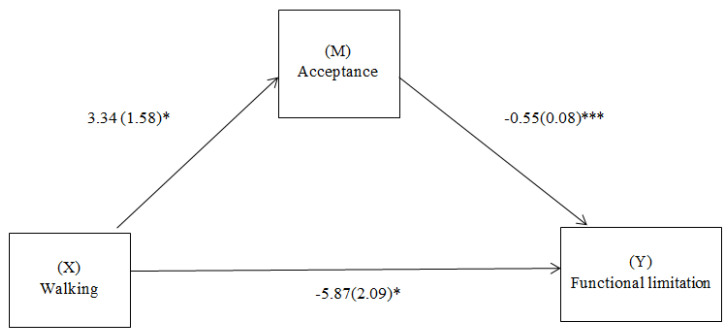
Path diagram illustrating the direct and mediating effects routes that relate walking with functional limitation (with acceptance as a mediator). Simple mediation analysis with walking as independent variable (X), functional limitation as dependent variable (Y), and acceptance as mediator (M). Values are unstandardized regression coefficients (SE in parentheses) and associated *p*-values (* *p* < 0.05, *** *p* < 0.001). Association in parentheses = direct effect (controlling for indirect effects). Solid lines indicate significant pathways and dashed lines indicate non-significant pathways.

**Figure 2 ijerph-20-05005-f002:**
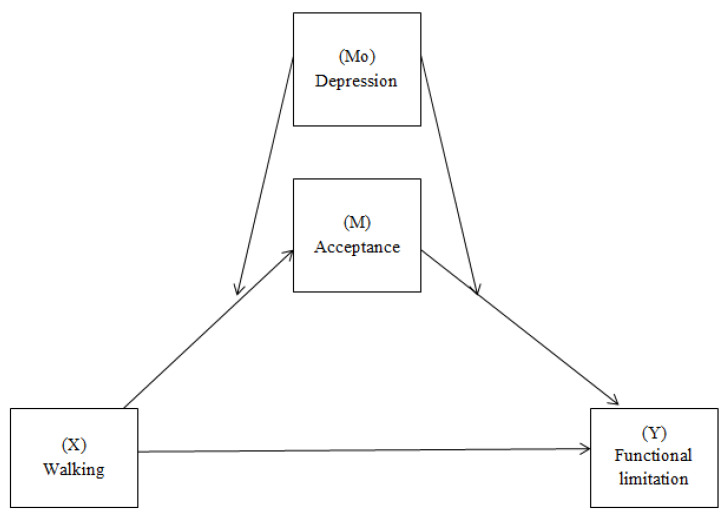
Diagram of the moderate mediation model with walking as independent variable (X), functional limitation as dependent variable (Y), acceptance as mediator (M) and depression as moderator (Mo) (Model 58).

**Table 1 ijerph-20-05005-t001:** Descriptives of psychosocial characteristics (*n* = 231).

Psychosocial Characteristics	Descriptives	Sample Range
Acceptance, mean (SD)	31.23 (11.80)	0–66
Functional limitation, mean (SD)	21.30 (5.71)	1–30
Depression, *n* (%) Yes No	90 (39.5) 138 (60.5)	
Walking, *n* (%)		
Yes	132 (57.9)	
No	96 (42.1)	

Abbreviations: SD (standard deviation); *n* (number); % (percentage).

**Table 2 ijerph-20-05005-t002:** Moderate mediation analysis (model 7).

	*R^2^*	*F*	*p*	Beta	*t*	*p*
Outcome variable = acceptance						
Model Summary	0.20	13.17	<0.001			
Walking				3.78	1.93	0.049
Depression				−6.56	−2.96	0.003
Walking × Depression				−5.91	3.01	0.049
Pain (covariate)				−0.02	−2.10	0.476
Outcome variable = functional limitation						
Model Summary	0.49	41.29	<0.001			
Walking				−2.46	−1.41	0.159
Acceptance				−0.42	−4.09	<0.001
Depression				4.86	0.98	0.327
Acceptance × Depression				0.02	0.13	0.899
Pain				5.82	10.57	<0.001

**Table 3 ijerph-20-05005-t003:** Indirect conditional effect at specific levels of the moderator when treating acceptance as a mediator.

Depression	Beta	SE	LL 95% CI	UL 95% CI
No	−2.05	1.11	−4.53	−0.22
Yes	0.77	0.98	−1.04	−2.80

Notes: SE = standard error; LL 95%CI = lower level of the 95% confidence interval; UL 95%CI = upper level of the 95% confidence interval.

## Data Availability

The data presented in this study are available on request from the corresponding author. The data are not publicly available due to privacy restrictions.

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
