# Peer review of "Does Pain Acceptance Contribute to Improved Functionality through Walking in Women with Fibromyalgia? Looking at Depressive Comorbidity"

_ijerph, 2023, doi:10.3390/ijerph20065005_

Round 1
Reviewer 1 Report
1) Abstract: Authors should reduce the “introduction” part, it is too long. Please include the “methods” part in the Abstract, how was the study been conducted, what study design, sampling method used, who were the participants etc. These info are missing in Abstract.
2) In your literature review, Is there any theory or empirical findings to support your model? Eg., why Acceptance is regarded as mediator in the model? Why depression as a moderator?
3) In your Discussion, Please explain why you don’t use structural equation modelling to study the association which involve mediator and moderator?
4) Any sample size calculation prior to the study? If yes, please include this information in your Method section.
Author Response
International Journal of Environmental Research and Public Health
Assistant Editor
Ms. Bailey Xu
Dear Bailey Xu,
We would like to thank you for your interest in our manuscript entitled “Benefits of walking on functional limitation in women with fibromyalgia. It is acceptance a strength in all cases? Looking at depressive comorbidity” (Reference ijerph-2219889). We appreciate the time that you and the other reviewers have dedicated to reading the manuscript and providing suggestions. Your suggestions have enriched the manuscript considerably. Likewise, we have incorporated all the comments suggested. Following your directions, we have proceeded to revise our manuscript, highlighting the changes by using the track changes mode in MS Word.
At the end of this letter, you will find an explanation of the changes made to the manuscript in accordance with your comments.
Once again, we wish to express our appreciation for the clear improvement of the article made possible by the reviewer and editor’s contributions. We hope the new changes meet their expectations, and we hope that they consider the work apt for publication in International Journal of Environmental Research and Public Health.
Please do not hesitate to suggest any further changes. We are at your disposal for anything else you may require.
Best regards,
1) Abstract: Authors should reduce the “introduction” part, it is too long. Please include the “methods” part in the Abstract, how was the study been conducted, what study design, sampling method used, who were the participants etc. These info are missing in Abstract.
Response: Thank you for your comment. The abstract has been modified based on the suggestions proposed.
2) In your literature review, Is there any theory or empirical findings to support your model? Eg., why Acceptance is regarded as mediator in the model? Why depression as a moderator?
Response: Thanks for your interesting comment. The proposed model is really based on two conceptual approaches: a) On the one hand, in relation to acceptance as a mediator, the application of the Acceptance and Commitment Therapy (ACT) in patients with chronic pain, shows the positive results of the same in the acceptance of their chronic condition and in the increase in functional autonomy (Vowles & Thompson, 2011). In this context, acceptance is a nuclear variable of the model. The process of acceptance of chronic pain is associated with a lower impact of the disease, including less disability (McCracken & Volwes, 2006; McCracken et al., 2004). Specifically, changes in acceptance, pain related anxiety, compensation strategies, and pain interference in walking ability has been found after the application of the ACT in chronic pain (Alonso-Fernández et al., 2016), b) on the other hand, proposing depression as a moderating variable in the model is due to the heterogeneity previously described in Fibromyalgia (Martínez et al., 2021) and the need to establish profiles or subgroups to respond to this reality and design more personalized treatments. Given the prevalence of depressive comorbidity in patients with fibromyalgia (Kleykamp et al., 2020), the analysis of this variable is proposed, establishing subgroups, incorporating depression as a moderating variable.
Following the reviewer's recommendations, this information has been incorporated into the introduction section of the manuscript.
Alonso-Fernández, M., López-López, A., Losada, A., González, J. L., & Wetherell, J. L. (2016). Acceptance and Commitment Therapy and Selective Optimization with Compensation for Institutionalized Older People with Chronic Pain. Pain Medicine (Malden, Mass.), 17(2), 264–277. https://doi.org/10.1111/pme.12885
Kleykamp, B.A., Ferguson, M.C., McNicol, E., Bixho, I., Arnold, L.M., Edwards, R.R., Fillingim, R., Grol-Prokopczyk, H., Turk, D.C., Dworkin, R.H. (2021). The Prevalence of Psychiatric and Chronic Pain Comorbidities in Fibromyalgia: an ACTTION systematic review. Semin. Arthritis Rheum, 51, 166–174, doi:10.1016/j.semarthrit.2020.10.006.
McCracken L M, & Volwes KE (2006). Acceptance of chronic pain. Curr Pain Headache Rep, 10:90–4. 31.
McCracken L M, Vowles K E, Eccleston C (2004). Acceptance of chronic pain: Component analysis and a revised assessment method. Pain, 107: 159–66.
Martínez, M.P., Sánchez, A.I., Prados, G., Lami, M.J., Villar, B., Miró, E. (2021). Fibromyalgia as a Heterogeneous Condition: Subgroups of Patients Based on Physical Symptoms and Cognitive-Affective Variables Related to Pain. Span. J. Psychol, 24, e33, doi:10.1017/SJP.2021.30.
Vowles K E, & Thompson M. (2011). Acceptance and commitment therapy for chronic pain. In McCracken LM, eds. Mindfulness and Acceptance in Behavioral Medicine: Current Theory and Practice. Oakland, CA: New Harbinger; 2011: 31–60.
3) In your Discussion, Please explain why you don’t use structural equation modelling to study the association which involve mediator and moderator?
Response: As Hayes mentions, in his text, there is no convincing justification for preferring analysis through PROCESS or structural equations. In any case, it is clear that PROCESS is an excellent instrument to test the effects of one variable on another in a study. In addition, the advantage of PROCESS is that it incorporates the bootstraping option that allows analysis with data that does not meet the assumptions of ordinary least squares.
Arcila Calderón, C. (2015). Avances metodológicos en los análisis de mediación, moderación y procesos condicionales. Universidad del Rosario/Universidad de Los Andes/Universidad Complutense de Madrid.
Hayes, A. (2005). Statistical Methods for Communication Science. Mahwah, New Jersey, Estados Unidos: Lawrence Erlbaum Associates.
Hayes, A.F., & Matthes, J. (2009). Computational Procedures for Probing Interactions in OLS and Logistic Regression: SPSS and SAS Implementations. Behavior Research Methods, 41(3), 924-936.
4) Any sample size calculation prior to the study? If yes, please include this information in your Method section.
Response: Yes, indeed the sample size was calculated. This information has been incorporated in the method section.

Reviewer 2 Report
Congratulations to the authors for the initiative to investigate in a population of women diagnosed with fibromyalgia because it makes the study very interesting from the current perspective of the active approach and patient-centered coping in chronic widespread pain.
PAGE 1. ABSTRACT
This should be revised and rewritten because it exceeds the maximum length of this section by more than 100 words.
PAGES 1-3. INTRODUCTION
The introduction gathers the most relevant literature and is well organized and structured, although it should be completed with references on central sensitization in patients with fibromyalgia, as it is a line of research in accordance with the objective and approach of this study, and with the conceptualization of kinesiophobia in patients with chronic widespread pain.
MATERIAL AND METHODS
PAGE 3
Lines 121-130.
What were the inclusion/exclusion criteria for selecting this sample? What was the sample universe? How was the calculation of the theoretical optimal sample size performed?
Who had made the diagnosis of the Fibromyalgia patients?
In addition to this dichotomous question, was the walking of the patients characterized in any way (number of days, walking time, intensity, distance covered,...)?
Line 137. Include the citation of the original questionnaire, not only of the Spanish validation.
Is the FIQ-R validated in Spanish?
How was the history of depression verified: general medical diagnosis, diagnosis by a psychiatric specialist, using a scale, etc.?
PAGE 4
Line 165. Was information collected on pharmacological treatment, medical history, main symptoms of the patients, family history, time of diagnosis of Fibromyalgia, ...?
RESULTS
PAGE 4
Line 183. Were there any losses or dropouts? Any missing data? Specify in this subsection.
DISCUSSION
PAGE 7
Line 273. As has been done with catastrophizing, the role of kinesiophobia should be discussed, as fear of movement influences the walking ability of Fibromyalgia patients.
PAGE 8
Within the psychological variables, positive mental health could be included in the discussion because of its influence on chronic widespread pain (fibromyalgia, chronic fatigue, ...).
What are the implications of this study for the care practice of patients diagnosed with fibromyalgia?
CONCLUSIONS
PAGE 8
Line 339. A conclusion in accordance with the secondary objective of the study would be missing.
Author Response
International Journal of Environmental Research and Public Health
Assistant Editor
Ms. Bailey Xu
Dear Bailey Xu,
We would like to thank you for your interest in our manuscript entitled “Benefits of walking on functional limitation in women with fibromyalgia. It is acceptance a strength in all cases? Looking at depressive comorbidity” (Reference ijerph-2219889). We appreciate the time that you and the other reviewers have dedicated to reading the manuscript and providing suggestions. Your suggestions have enriched the manuscript considerably. Likewise, we have incorporated all the comments suggested. Following your directions, we have proceeded to revise our manuscript, highlighting the changes by using the track changes mode in MS Word.
At the end of this letter, you will find an explanation of the changes made to the manuscript in accordance with your comments.
Once again, we wish to express our appreciation for the clear improvement of the article made possible by the reviewer and editor’s contributions. We hope the new changes meet their expectations, and we hope that they consider the work apt for publication in International Journal of Environmental Research and Public Health.
Please do not hesitate to suggest any further changes. We are at your disposal for anything else you may require.
Best regards,
PAGE 1. ABSTRACT
This should be revised and rewritten because it exceeds the maximum length of this section by more than 100 words.
Response: Thanks for your suggestion. This section has been reduced.
PAGES 1-3. INTRODUCTION
The introduction gathers the most relevant literature and is well organized and structured, although it should be completed with references on central sensitization in patients with fibromyalgia, as it is a line of research in accordance with the objective and approach of this study, and with the conceptualization of kinesiophobia in patients with chronic widespread pain.
Response: Thank you very much for your suggestions. The requested information has been included.
MATERIAL AND METHODS
PAGE 3
Lines 121-130.
What were the inclusion/exclusion criteria for selecting this sample? What was the sample universe? How was the calculation of the theoretical optimal sample size performed?
Response: The information requested in the procedure section has been incorporated.
Who had made the diagnosis of the Fibromyalgia patients?
Response: The information requested in the procedure section has been incorporated. The diagnosis was made by the rheumatology or primary care service.
In addition to this dichotomous question, was the walking of the patients characterized in any way (number of days, walking time, intensity, distance covered,...)?
Response: No additional question was asked because the wording of the question specified the specific guideline that should have been carried out. This is: “walk 2 to 4 days a week, a minimum of 30 min a day, in 15-20 min shifts, with a small rest between shifts for a minimum of six consecutive weeks in order to exercise”
Line 137. Include the citation of the original questionnaire, not only of the Spanish validation.
Response: The quote from the original questionnaire has been incorporated. Additionally, thanks to the reviewer's comment, all the instruments have been reviewed, proceeding to incorporate both the original version and the Spanish validation used in this study.
Is the FIQ-R validated in Spanish?
Response: Yes, the FIQ-R is validated in Spanish. The validation reference has been added.
How was the history of depression verified: general medical diagnosis, diagnosis by a psychiatric specialist, using a scale, etc.?
Response: Thank you for your comment. The diagnosis of depression was verified through the clinical history of the patients. Additionally, the Hospital Anxiety and Depression Scale (HADS) (Herrero et al., 2003; Zigmond et al., 1983) (depression dimension) was administered, verifying that the patients with a diagnosis of depression had scores equal to or greater than 12 on the depression scale, in accordance with previous literature validating the instrument in Spain in patients with fibromyalgia (Cabrera et al., 2015). This information has been incorporated in the Method section.
Cabrera, V.; Martín-Aragón, M.; Terol, M. del C.; Núñez, R.; Pastor, M. de los Á. La Escala de Ansiedad y Depresión Hospitalaria (HAD) en fibromialgia: Análisis de sensibilidad y especificidad. Ter. psicológica 2015, 33, 181–193, doi:10.4067/S0718-48082015000300003
Herrero, M.J.; Blanch, J.; Peri, J.M.; De Pablo, J.; Pintor, L.; Bulbena, A. A validation study of the hospital anxiety and depression scale (HADS) in a Spanish population. Gen. Hosp. Psychiatry 2003, 25, 277–283, doi:10.1016/S0163-8343(03)00043-4.
Zigmond, A.S.; Snaith, R.P. The Hospital Anxiety and Depression Scale. Acta Psychiatr. Scand. 1983, 67, 361–370, doi:10.1111/j.1600-0447.1983.tb09716.x.
PAGE 4
Line 165. Was information collected on pharmacological treatment, medical history, main symptoms of the patients, family history, time of diagnosis of Fibromyalgia, ...?
Response: Thanks for your suggestion. The information on the pharmacological treatment and the time of diagnosis of fibromyalgia has been incorporated. In relation to the main symptoms of the patients, these were measured through the FIQ-R, whose “Descriptive Analysis” appear in the manuscript because they constitute one of the variables under study of the proposed model. However, data related to family and medical history were not available.
RESULTS
PAGE 4
Line 183. Were there any losses or dropouts? Any missing data? Specify in this subsection.
Response: Thank you for your comment. The required information has been incorporated into the manuscript. Initially, 268 participants agreed to participate in the study. However, 22 patients did not attend the scheduled evaluation appointment, 6 patients did not sign the informed consent and the questionnaires of 9 patients contained a large amount of missing data, so it was decided to eliminate them from the study. For these reasons, the final sample consisted of 231 participants.
DISCUSSION
PAGE 7
Line 273. As has been done with catastrophizing, the role of kinesiophobia should be discussed, as fear of movement influences the walking ability of Fibromyalgia patients.
Response: The requested information has been incorporated.
PAGE 8
Within the psychological variables, positive mental health could be included in the discussion because of its influence on chronic widespread pain (fibromyalgia, chronic fatigue, ...).
Response: Thanks for your suggestion. Indeed, as the reviewer points out, acceptance is analyzed as a positive variable associated with mental health. New references have been added about the influence of positive mental health on chronic pain.
Hassett, A.L.; Simonelli, L.E.; Radvanski, D.C.; Buyske, S.; Savage, S. V.; Sigal, L.H. The relationship between affect balance style and clinical outcomes in fibromyalgia. Arthritis Rheum. 2008, 59, 833–840, doi:10.1002/art.23708.
Zautra, A.J.; Fasman, R.; Reich, J.W.; Harakas, P.; Johnson, L.M.; Olmsted, M.E.; Davis, M.C. Fibromyalgia: Evidence for Deficits in Positive Affect Regulation. Psychosom. Med. 2005, 67, 147–155, doi:10.1097/01.psy.0000146328.52009.23.
Segura-Jiménez, V.; Estévez-López, F.; Soriano-Maldonado, A.; Álvarez-Gallardo, I.C.; Delgado-Fernández, M.; Ruiz, J.R.; Aparicio, V.A. Gender Differences in Symptoms, Health-Related Quality of Life, Sleep Quality, Mental Health, Cognitive Performance, Pain-Cognition, and Positive Health in Spanish Fibromyalgia Individuals: The Al-Ándalus Project. Pain Res. Manag. 2016, 2016, 1–14, doi:10.1155/2016/5135176
What are the implications of this study for the care practice of patients diagnosed with fibromyalgia?
Response: Thanks. Information on practical implications in the treatment of fibromyalgia patients is listed at the end of the discussion.
“These findings have important practical implications as they suggest that implementation-oriented treatments for walking in fibromyalgia do not have the same effect in patients with depression. In this last case, a priori, walking would not have a positive impact on the functionality of patients with fibromyalgia. Thus, working in the first place to reduce the levels of depression in these patients would be the most appropriate treatment. These results are consistent with previous studies showing that the benefits of leading an active lifestyle vary depending on the characteristics of the population examined [12,63,64]. Our results confirm that, before promoting the pre-registration of aerobic physical exercise (e.g. walking) in order to increase the functionality of these patients, it would be necessary to carry out a previous analysis of the levels of depression”.
CONCLUSIONS
PAGE 8
Line 339. A conclusion in accordance with the secondary objective of the study would be missing.
Response: Thanks for your suggestion. A conclusion according to the second study has been incorporated.
